# The Influence of Social Media on Perceived Levels of National Security and Crisis: A Case Study of Youth in the United Arab Emirates

**Nadir Al Naqbi [1], Naill Al Momani [1] and Amanda Davies [2],***

1 Emergency Management Department, Dubai Police Academy, P.O. Box 1493, Dubai, United Arab Emirates
2 Policing and Security, Rabdan Academy, P.O. Box 114646, Abu Dhabi, United Arab Emirates
* Correspondence: adavies@ra.ac.ae

**Abstract:** The increase in the use of social media as a 21st century communication tool is in parallel increasing the threat to national security globally. This study explores the perception of United Arab Emirate community members (specifically youth) on the influence of social media as a threat; the wide use of SM platforms for Emirate of Sharjah (Dibba Al-Hisn, Khor Fakkan, Kalba) were analyzed utilizing a descriptive-analytical method. The results of the study on the effects and consequences of social media on national security in the UAE, rates social media as having the highest level of influence on political implications followed in decreasing order of influence by, economic, cultural and societal, ethical and religious dimensions, and the least potential influence being on perceived national security implications. Further, the results of a one-way variance analysis indicate the potential for the perceived level of national security experienced by youth community members in the UAE to be predicted through social media. A unique feature of this study is the analysis of the influence of the five dimensions of national security on each other and national security collectively from an Arab youth perspective. Further, the study design is replicable and offers, (a) an opportunity for wider utilization as an avenue for contributing to understanding the impact of social media on the perception of a country's national security, and (b) a fundamental baseline for future research.

**Keywords:** national security; social media; youth; crisis; United Arab Emirates; Emirate of Sharjah; multiple linear regression; social media influence; Arab youth

## 1. Introduction

Security is, according to Maslow's Hierarchy of Needs theory [1], a fundamental need of human life and the level to which this is experienced by members of a community is reflected in the confidence and security of the respective society. As governance bodies across the world seek to establish sustainable solutions to the provision of national security and disaster management for their communities, they are challenged in addressing the conundrum of managing the influence of social media.

Social Media (SM), its reach, impact, and potential in a globalized world are no longer contested; it has affected people's lives, regarding its use and misuse [2]. The phenomenal rise in the use of the internet with associated simple and multi-layered social media platforms is recognized globally as one of the most significant threats to security at the local, regional, national, and global level. The role of social media as a mechanism for seeking to destabilize intellectual convictions, ideological constants, and moral and social virtues to create an imbalance within society, increasing the threat to national security and crisis, is widely acknowledged in current literature [3–5]. The research presented in this paper seeks to inform on the influence of social media on elements contributing to national security as experienced in the United Arab Emirates. Al-Huwaish [6] suggests security comprises different elements and their priority is different for each individual. These security elements include social, political, economic, criminal, food, water, cultural, environmental, and

information security (environmental and information being more recent security additions), all of which are interdependent. Al Huwaish [6] further proffers the interdependence of these elements of security results in disruption across the comprehensive set of security when one of the elements is lost or disrupted.

The current study explores two key questions: What is the impact of social media as one of the new media tools, on national security from the perspective of young people? Second, is it possible to predict the level of national security in the UAE through the impacts of social media on society from the perspective of young people? The findings from this study exploring the perspective of UAE youth regarding the influence of social media on key domains, including political, cultural, economic, security, societal, ethical and religious domains, contributes to (a) expanding the knowledge of the influence of social media through the perceptions of Arabic youth in relation to a nations' level of security; and (b) considerations for policy and legislation in regard to a nations' social media platform operations internally and externally. In parallel with developing an understanding of the perception of youth on the influence of social media on national security is the contribution of the study to the next step, addressing the influence, or as Tawfiq [7] suggests, mitigation of the negative attributes of social media.

## 2. Background

In the context of this study, the suggestion by Al-Huwaish [6] is that the makeup of national security comprises interdependent elements including social, political, economic, criminal, food, water, cultural, environmental, and information security (environmental and information being more recent security additions). The work of Al-Suwaidi [8] argues that the 21st century communication tool, the internet, and associated media platforms, and its excessive use in society, especially by youths, has had repercussions/impacts on these various social, economic, political, intellectual, ideological, moral, personal, and national security levels within the Arab world. The work of Norri-Sederholm et al. [9] suggests social media is an increasing security threat impacting personal lives and a key channel for dissemination of information for political gain. This influence, Norri-Sederholm et al. [9] suggest, is of particular significance for youth as the internet has become:

> . . . a core part of their everyday reality and not just a peripheral place to visit and share ideas with others. Young people's connections and networks in social media also provide them with the opportunity to engage in different types of political discussions in society. (p. 231)

Al Zaabi and Tomic [10] in their work researching the influence of the use of social media platforms on recruiting and supporting terrorist activities conclude that there is evidence that social media can be directly linked to terrorism, planning, recruitment, and attacks. The authors suggest the overall outcome of the use of social media platforms has resulted in successful recruitment campaigns, particularly in youths, to join terrorist groups. Similarly, the work of Al-Saggaf and Davies [11] in examining the expression of grievances in the Arabic Twitter sphere identified the willing engagement with these topics on online social media platforms. The implication here is the direct influence of the internet and associated platforms to disseminate influential campaigns to disrupt the security of people's lives and the future of their national security. Vasu et al., [12] in reporting on the influence of fake news and referring to the velocity of information, which is spread within seconds online, suggests this form of media offers those seeking to destabilize a state, to readily, and with impact, spread disinformation and achieve their aims. Al-Enzi [13] explains that social media has become one of the most important tools on which terrorist and criminal groups rely in spreading misleading rumors and destructive ideas. The work of Al Zaabi and Tomic [10] advocates for research that identifies the means for controlling terrorist activities on social media platforms:

> ... Any future researcher must therefore understand that 90% of terrorist activities have gone dark, meaning that it is high time to research the strategies to control their clandestine activities. (p. 1)

Similarly, Cardenas et al. [14] support the urgency of research to identify behavior bordering on criminality within the deregulated world of social media as a human security imperative for governments (p. 1). This call for further research is an underpinning premise for this current study, that is to develop a more in-depth understanding of the influence of social media on the perceived level of national security as seen through an Arabic youth lens. Specifically, the study has been undertaken with Arabic youth in the United Arab Emirates.

There is a rapidly developing body of literature which collectively indicates that social media has become the communication tool of choice and, either by default or intent (dependent on the user), is influencing peoples' perspectives on personal and national issues [9,11]. Social media has altered the way we become informed and form opinions. The role of online social media for influencing the perception of national security of a nation from an external view is discussed in the 2020 work of Marine-Roig [15], analyzing online travel reviews and the use of electronic word-of-mouth. Marine-Roig's [15] work centered on analyzing the impact of online social media tourism-related platforms following a period of terrorist-related activities and unrest in Catalonia. The work suggests:

> ... results show that serious events had a minimal impact on the city's image as perceived and shared by reviewers despite the enormous media coverage.

An additional work of Marine-Roig [16] and Marine-Roig and Huertas [17] explores the influence of user-generated content (UGC) suggesting such content shared on social media has the capacity to impact the online image of a country and the ranking as a tourism destination, and by association perceptions of safety and security of visitors and citizens. Similarly, the work of Tsoy & Tirasawasdichai [18] analyzing the role of social media in shaping public risk perception during the COVID-19 pandemic suggests:

> ... while traditional media still plays a significant role in shaping risk perception, social media can be considered even more influential. (p. 39)

Whilst there is now more recent work published in relation to the influence of social media on society, the earlier work of Al Sumadi [19] in exploring the role of social networks/media sites (Facebook and Twitter and WhatsApp and YouTube) signposts the strong potential influence of these communication tools for disrupting the perception of security and, specifically in the Al Sumadi [19] study, intellectual security. Al Sumadi [19] defines intellectual security as:

> ... means to maintain authentic cultural components to face other foreign cultural currents that may be suspicious. (p. 633)

According to Al Sumadi [19], a review of work in this field in the period leading up to 2016 was limited, explaining:

> ... we should point to the rarity of studies that deal directly with the effect of social networks causing Intellectual Deviation. (p. 636)

The dimension of Intellectual Deviation as explained by Al Sumadi [19] resonates with the current UAE study through the association of culture and religion; however, a similar situation occurs as experienced by Al Sumadi [19] in exploring the social network/media influence on Intellectual Deviation, in that there is no direct comparison in terms of the survey dimensions of the UAE. The UAE study does enable insight into a broader range of dimensions on national security. Al Sumadi [19] concludes that social networking/media does cause intellectual deviation through the potential to use the platforms to spread social, political, and religious ideas. The work of Stanger et al. [20] encompassed a different angle from which to explore the relationship between social networks/media and Arabic culture and religion. In the Stanger et al. [20] work, the survey instrument sought to

understand how those with an Arabic cultural and religious background used social media. Although this study does not directly align to the current UAE study, the topic of this paper, the findings, and the results resonate with the UAE results suggesting limited negative influence of social media experienced on cultural and religious perspectives.

A central tenet of this study is the concept of social media as a fertile environment that some exploit to exchange misconceptions among its users to create internal and external national instability, i.e., the systematic distribution of communications via social media to gain support from young people to impact on the national security of the state in many areas (cultural and societal, moral and religious, political, economic, security). A review of the literature associated with the Arab Spring suggests social media played a major role in sabotaging and causing unrest in several Arab countries in the recent period in the so-named 'Arab Spring' protests. Terrorist satellite channels exploited their various social platforms to incite people against their governments whilst others used it to distribute misinformation about the UAE, the aim of which was to destabilize UAE national security and internal stability [21–24]. The spread of these networks and websites in the UAE society, due to their technological and educational potential, poses a major challenge because of their negative and positive impact on society, and at all political, economic, educational, cultural, social, security, religious, and moral levels.

Globally, there has been an increasing body of literature focused on understanding the impact of social media across multiple domains associated with youth, chief amongst which are mental health wellbeing, for example [25–27], and political activism, for example [28–31]. The literature does not definitively confirm the use of social media by youth as positive or negative, rather there are elements of both and it may be influenced by the lens through which the research is conducted. Importantly, in respect of this study, as suggested by Auxier and Anderson in 2021 [32], the youth category is the most frequent user of social networking sites.

Saleh [33] suggests social media has added positive dimensions to people's lives, and has promoted social, cultural, scientific, and political changes in their lives, including elements as briefly described as follows. Self-enhancement whereby social media have contributed to the self-enhancement of individuals by giving them the opportunity to create an independent entity in the community in which they express themselves. Social media has expanded communication networks, suggesting social media has contributed to the exchange of common interests, habits, and activities. Hence, individuals could search for friends with common hobbies and qualities around the world. Additional positive aspects suggested by Saleh [33] include social media as a platform for the exchange of ideas and the expression of personality, intellect, culture, and beliefs of the participant. Enabling connection with family and friends across the world and connecting businesses have been identified as further positive attributes of social media.

In balance, Al-Labban [34] identified a set of negative attributes of social media, and although not an exhaustive list, the following offers an insight into the broad areas of negative influence. These areas resonate in the literature exploring the impact of social media more widely, for example: the work of Alexander [35]; Akram & Kumar [36]; Raggad & Shweihat [37]. The areas proffered by Al-Labban [34] include the addictive nature of social media whereby it generates circumstances where the user loses track of time and place and connection within the reality of the lived moment. One of the more dangerous influences from a national security perspective is the opportunity in social media to hide one's true identify, the ability to use pseudonyms. Current literature supports the suggestion by Al-Labban [34] of the negative influence of social media on the perception of national security through dissemination of destructive ideas and the spreading of a negative culture, e.g., violence and terrorism. Research exploring the role of social media in the support of terrorist-related activities is widely published and the work resonates with the suggestions of Al-Labban [34], Hollewell and Longpre [38], KhosraviNik and Amer [39], and Cherney et al. [40]. In the context of this study, the work of Al-Labban [33]

suggests social media supports the loss of Arab culture identity replacing it with a more global identity.

In the context of establishing and sustaining national security, Morsi [41] indicates achievement requires: (a) recognizing the threats and challenges and potential risks; (b) use of soft and military power to defend national security; (c) provide the capacity to defend national security; and (d) utilizing future foresight data to build scenarios to prepare for future threats. The relationship between these requirements and social media usage lies in understanding the impact of social media on (a) and (b). In addition to the negative aspects of social media discussed above, the work of Al-Huwaish [6] highlights the governance challenges associated with social media, suggesting an additional threat to national security is enabled where the platforms/channels of social media are not subject to frameworks or laws that control the content disseminated and circulated through the respective social media platforms.

The study presented here included exploring the relative level of influence of social media on religion and through religion, the perceived level of national security experienced by youth in the UAE. In association with the impact on religion, Abdul-Khalequ [42] suggests social media has enabled groups bearing characteristics of Islam to communicate and campaign ideas that are not reflective of authentic Arab values, for example the actions of ISIS in Iraq and Syria in 2018 in recruitment activities. The work of Al-Smadi, [19] and Acar [43], similarly suggests that the role of social media has provided an opportunity to influence cultural beliefs and ethical standards and actions. As discussed by McElreath [44], and Huda et al. [45], social media has become a vehicle for extremist organizations to document, broadcast, plan, and carry out their terrorist operations. The social media platforms enable members of terrorist organizations to use them to communicate with each other, to exchange private information, to spread it in remote geographical areas, and to relate them organizationally or intellectually to each other. This perspective is similarly reflected in the work of Akram and Kumar [36] in proffering that social media plays a central and strategic role in influencing the values of citizenship and loyalty, and it is a means of dialogue and marketing the image of the state, its values and its civilization to the world, and psychologically influencing followers, particularly youth.

Developing an understanding of the perspective Arabic youth have on the influence of social media on cultural, economic, political, religious, and security domains through this current study is unique. The point of difference is rather than isolate the individual elements, this research has sought to build knowledge of the perception of influence on the combination of aspects and the relationship between each element. A further point of difference to previous research is to explore in this research the potential for the perception of influence of social media to predict the level of national security perceived by youth in the UAE. Figure 1 offers a conceptual diagram of the relationship between the five elements and the aims of the research.

A significant challenge with the current UAE study is the limited available literature reporting similarly comparable studies. This situation is due in part to the orientation of studies associated with the influence of social media in society. As indicated in Table 1, whilst by no means an exhaustive list of studies, in the domain of social media and influence in society, the examples are indicative of the variation in the focus of studies on the influence of social media. One of the noteworthy aspects of the UAE study, is what it is not; it does not attempt at this stage to explore the nuances within the dimensions, in particular, as there is a wealth of knowledge in this area, and it does not delve into social media influence on radicalization and terrorism.

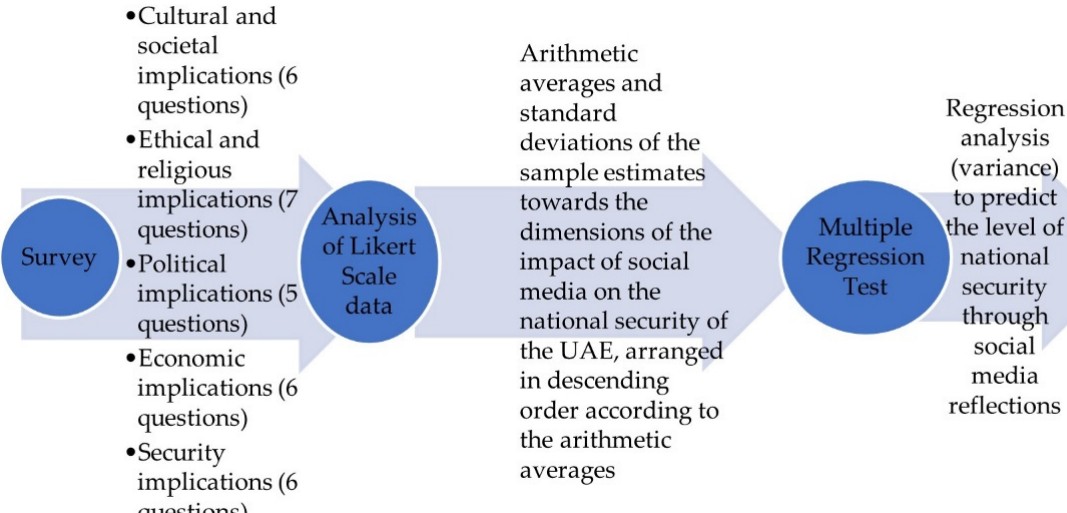

**Figure 1.** Conceptual diagram of the relationship between the five elements (survey) data and the aims of the research.

**Table 1.** Comparative literature.

| Author/s | Reference Title | Content Focus | Current Study Comparison/Contribution |
|---|---|---|---|
| Norri-Sederholm, T., Norvanto, E., Talvitie-Lamberg, K., Aki-Mauri Huhtinen, A. [9] | Social Media as the Pulse of National Security Threats: A Framework for Studying How Social Media Influences Young People's Safety and Security Situation | A four-year study not yet complete, the study approaches the subject from the perspective of society's comprehensive security and investigates whether activities in social media influence attitudes towards personal and national security, and young people's safety and security situation. | Findings not yet available from the Finnish study, the current UAE study will offer a valuable comparison; however, there may be a difference in the dimensions investigated in the two studies. |
| Al Zaabi, K., Tomic, D. [10] | New security paradigm—the use of social networks as a form of threat to the national security state | This qualitative study examined the role of social media in influencing indoctrination. | The study design is not directly comparable to the UAE study; the Al Zaabi and Tomic study suggests social media is a tool for influencing national security—the UAE study indicates there is limited impact. |
| Al-Enezi, N. N. [13] | Employment of social networking sites in response to rumors | Study exclusively explored the role of social media managing Facebook so as to mitigate false information. | The study confirmed the wide use of social media by youth and the potential of its influence. The UAE study resonates the findings. |
| Marine-Roig, E. [15] | Content analysis of online travel reviews | Study explores online travel reviews—specifically related to perspective on image of areas with recent terrorist activity. | The study aligns with the UAE findings that social media activity had limited impact on the perspective of security in specific locations. |

**Table 1.** *Cont.*

| Author/s | Reference Title | Content Focus | Current Study Comparison/Contribution |
|---|---|---|---|
| Tsoy, D., Tirasawasdichai, T., & Kurpayanidi, K. I. [18] | Role of social media in shaping public risk perception during COVID-19 pandemic: | This study did not contain data, it was a review of literature, the theme of which indicates social media exposes people to more information and the potential to heighten risk perception, calling for attention to crisis communication management and the use of social media to examine public opinion. | There is no comparable data, the study is valuable in confirming the strength of social media and its potential to heighten risk perception and by default perception of security. |
| Akram, W., Kumar, R. [36] | A study on positive and negative effects of social media on society | A presentation of positive and negative effects of social media on society—no data available—a general review of commonly used social media sites and the positive and negative effects on education, business, society generally, teens, and children. | The study does not offer any data with which to establish a comparison, the study does support the potential of social media to have positive and negative effects on these dimensions in society.The UAE study offers an extension to this study through provision of data and analysis of youth use of social media. |
| Raggad, A., & Shweihat, S. [37] | The degree of positive and negative effects of social media networks from the point of view of the German-Jordanian University students | The study includes a questionnaire of 55 paragraphs and analyses the most attractive topics on social media from the perspective of a sample of German-Jordanian University students. The current study aimed to know the reality of the use of social networks by students of the German-Jordanian University in terms of the topics they follow and the sites that are most attractive to them, analyzing the social and cultural effects (negative and positive). | The approach in the Raggad and Shweihat is similar to this current UAE study, the focus of the questions is different in that the UAE study is exploring specifically the connection between the influence of social media on perspectives of national security, the Raggad and Shweihat study explores which topics on social media are of most interest and the students' perspectives of the positive and negative influences of social media related to those topics. |
| Hollewell, G.F., Longpre, N. [38] | Radicalization in the social media era: Understanding the relationship between self-radicalization and the internet | This study focused on the role of social media in self-radicalization. Results showed that individuals holding a university degree—especially young men—were more at risk of endorsing positive attitudes toward political violence and terrorism, and, therefore, more at risk of being radicalized. | The Hollewell and Longpre study questions focused on understanding the emotional intelligence, psychological involvement on social media, attitudes toward terrorism, and political violence, and loneliness. These dimensions did not directly align with the focus of the UAE survey dimensions. |

**Table 1.** *Cont.*

| Author/s | Reference Title | Content Focus | Current Study Comparison/Contribution |
|---|---|---|---|
| Al Smadi, H. [19] | The Effect of Social Networking Sites In Causing Intellectual Deviation From Qassim University Students' Perspective | The study focusses on investigating the effect of social networks in causing intellectual deviation (distortion of Islam and noble values) by KSA university students. | The study utilized a questionnaire and analyzed the results in a similar process to the UAE study—means & standard deviation; Pearson correlation to treat variables, Manova analysis. The results suggest a potential for a strong influence by social media on intellectual deviation (culture/religion/values of Islam). The study does not consider national security. The difference here with the UAE study results suggest the core dimensions are interdependent with security least impacted. |
| Stanger, N., Alnaghaimshi, N., Pearson, E. [20] | How do Saudi youth engage with social media | The study utilizes Hofstede's cultural dimensions to assess how cultural and religious factors are shaping the use of social media (Instragram, facebook, snapchat). The research sample used was KSA students studying in New Zealand. | Surveys and interviews were conducted, the results indicating the sample were very conscious of behaving ethically and culturally and religiously appropriately on social media. These were not the dimensions of the UAE study; however, the results do resonate with the UAE results suggesting limited negative influence of social media experienced on cultural and religious perspectives. |

## 3. Materials and Methods

The study falls within the case study research design as discussed by Yin [45]. Yin [45] suggests case studies are utilised when a researcher is studying specific contextual conditions to understand more extensively the phenomenon under study. In this specific case, it is the study of the response by youth as to their perception of the influence of social media on national security. The current study does not readily align with the development of a hypothesis, which would, according to Yin [43], be invaluable for a case study approach. However, the data collection of utilising a survey supports the case study approach, particularly as the strategy aligns with the definition of a case study by Robert Burns [46]. Burns [46] advocates a case study would typically include observation of a specific aspect/unity under study.

This study relied on the descriptive analytical method and a questionnaire as a tool for data collection. The questionnaire consisted of a set of questions in two categories. Category One centered on understanding the perception of the implications of social media across five societal dimensions:

- Cultural and societal implications (6 questions);
- Ethical and religious implications (7 questions);
- Political implications (5 questions);
- Economic implications (6 questions);
- Security implications (6 questions).

Furthermore,

Category Two centered on understanding the perception of the overall level of national security viewed through the lens of youth in the UAE. The complete set of survey questions is presented at Appendix A.

The questions were designed on the Likert Scale format offering response selections of Strongly Agree (5), Agree (4), Neutral (3), Disagree (2), and Strongly Disagree (1). As discussed by Burns [46], the advantages of the Likert Scale are:

> ... the fact that the method is based entirely on empirical data regarding subjects' responses rather than subjective opinions of judges; and the fact that this method produces more homogeneous scales and increased the probability that unitary attitude is being measured, therefore that validity (construct and concurrent) and reliability are reasonably high. (p. 560)

The study potential population consisted of all young people in the cities of the Eastern Region of the Emirate of Sharjah (Dibba Al-Hosn City, Khor Fakkan City, Kalba City), and their total number is approximately 19,600, according to the data of the Department of Statistics and Community Development in Sharjah for the year 2020. The study sample consisted of (1185) young people (male/female) who were chosen by a simple random method. The reason for determining the age group of youth in this study (20–40 years) was based on the levels and age groups approved by the Department of Statistics and Community Development in the Emirate of Sharjah, and potentially the group indicating high usage of social media in the UAE. As the 13th Annual ASDA'A BCW Arab Youth Survey released in October 2021 indicates, almost two-thirds (61%) of young Arabs (aged 18 to 24) use social media for news [47]. The averages and standard deviations of the study sample estimates were extracted from the questions pertaining to the implications of social media on national security in the UAE. The arithmetic averages and standard deviations are highly regarded as reliable technical indicators used in data analysis and in describing trends accurately [48,49].

## 4. Results

The results attributed to the first question of the study are presented in Table 2: What are the implications of social media on national security in the UAE from the point of view of the study sample?

**Table 2.** Arithmetic averages and standard deviations of the sample estimates towards the dimensions of the impact of social media on the national security of the UAE, arranged in descending order according to the arithmetic averages.

| Rank: | Dimension No. | Dimension Descriptor | Arithmetic Mean | Standard Deviation | Grade Degree |
|---|---|---|---|---|---|
| 1 | 3 | Political implications | 3.46 | 0.46 | High |
| 2 | 4 | Economic implications | 2.92 | 0.39 | Moderate |
| 3 | 1 | Cultural and societal implications | 2.79 | 0.40 | Moderate |
| 4 | 2 | Ethical and religious implications | 2.35 | 0.55 | Low |
| 5 | 5 | Security Implications | 2.13 | 0.67 | Low |
| | Overall Scale | | 2.69 | 0.38 | Moderate |

Table 2 illustrates the arithmetic averages and standard deviations of the study sample's estimations about the implications of social media on the perception of levels of national security in the UAE. Ranked in first place and therefore deemed to be the highest degree of influence in this study is Political implications, as illustrated in Table 2. Economic implications ranked second highest (mean = 2.92; standard deviation = 0.39). Cultural and societal implications (mean = 2.79; standard deviation 0.40) ranked third. The Ethical and religious implications dimension ranked fourth (mean = 2.35; standard deviation 0.55). The dimension ranking lowest in this study was Security implications (mean = 2.13; standard deviation = 0.67).

The results presented in Table 2 indicate there is a low-level difference in the dimensions influencing the perception of national security through social media in this study.

In respect of the second research question, "Can the level of national security in the UAE be predicted through the implications of social media based on the study sample?", the multiple regression test was applied. All the independent variables were entered into the multiple linear regression equation in order to find out which dimensions can be predicted by the level of national security in the UAE through the reflections of social media. Table 3 presents the analysis of the multiple linear regression equation applied to the data.

**Table 3.** Regression analysis (variance) to predict the level of national security through social media reflections.

|  | Sum of Squares | Degrees of Freedom | Mean Square | Value (F) | Statistical Significance |
|---|---|---|---|---|---|
| Regression | 10.175 | 5 | 2.035 | 55.431 | 0.000 |
| Residual | 43.282 | 1179 | 0.037 |  |  |
| Total | 53.457 | 1184 |  |  |  |

The data suggests there is a level of indicative influence from the five dimensions (Regression) and a substantial (43.282) level of influence from other sources—it is, however, not possible in this study to determine which dimensions comprised the 'Residual'. Table 4 presents the results of the analysis of the regression equation coefficients.

**Table 4.** Results of multiple regression analysis to predict the level of national security in the United Arab Emirates through the implications of social media.

| Independent Variables | Regression Coefficient | Standard Error | Beta Coefficient (B) | Value (T) | Statistical Significance |
|---|---|---|---|---|---|
| (Constant) | 2.907 | 0.058 |  | 50.000 | 0.000 |
| Cultural and societal implications | 191 | 0.017 | 0.363 | 11.551 | 0.000 |
| Ethical and religious implications | −0.099 | 0.016 | −0.259 | −6.151 | 0.000 |
| Political implications | 0.020 | 0.013 | 0.044 | 1.509 | 0.132 |
| Economic implications | 0.037 | 0.017 | 0.069 | 2.157 | 0.031 |
| Security Implications | −0.090 | 0.014 | −0.288 | −6.674 | 0.000 |
| Dependent variable: National Security | | | | | |
| Correlation coefficient R = 0.436 | | | | | |
| Coefficient of determination $R^2$ = 0.190 | | | | | |
| Explained variance = 0.187 | | | | | |

It is noted that the statistical significance in the following dimensions—cultural, societal, moral, religious, economic, and security implications, is less than the significance level (0.05); which means that these dimensions can be predicted to increase or weaken the perception of the level of national security in the United Arab Emirates from the viewpoint of UAE youth.

The value of the correlation coefficient (R) (0.436) indicates that the dimensions of cultural, societal, moral, religious, economic, and security implications indicate a weak correlation between them and the level of national security in the United Arab Emirates. While the square of the correlation coefficient ($R^2$) was 0.190, the percentage of the variance that is explained by the dimensions of the combined social media implications was 18.7%, with the remaining 81.3% attributed to other variables which did not enter the multiple linear regression model.

It is also noted that the values of the regression coefficient and the beta coefficient (B) were positive in the following two dimensions: cultural and societal implications, and economic implications, which indicates that these two dimensions directly affect the national security. Accordingly, a 1% change in the level of the two dimensions (cultural and societal implications; economic implications) causes a change in the level of national security by 36.6% and 6.9%, respectively. While the values of the regression coefficient and beta coefficient (B) were negative in the following two dimensions: moral and religious implications, and security implications, this indicates that these two dimensions negatively affect the level of national security. Accordingly, a change of 1% in the level of the two dimensions (ethical and religious implications; security implications) leads to a change in the level of national security by 25.9% and 28.8%, respectively. In addition, the regression equation for the dimensions of the social media implications combined are predictable through the application of the following equation: (national security level = 2.907 + (0.191 × cultural and societal implications) − (0.099 × moral and religious implications) + (0.037 × economic implications) − (0.090 × security implications).

## 5. Discussion

Social media is a double-edged sword. It is positive in that it facilitates communication between individuals with common interests and hobbies, relatives or friends, and quickly exchanges information. It is also a fertile environment for exchanging ideas and information, learning about cultures, and the speed of access to news and events at both national and global levels. On the other hand, its negative aspect is its use to destabilize the security and stability of communities/society at national and international levels. As social media has become an integral part of modern society with 58.7% of the world's population using social media [49], it has the potential to affect various social, economic, political, intellectual, ideological, moral, and security levels. An interesting finding in this study is the ranking of SM influence on security, which was situated as the least influenced. This resonates with the findings in the work of Marine-Roig [15] which indicated that despite significant social media coverage, there was limited negative impact on tourism in an area of recent terrorist activity. There does need to be a measure of caution in the interpretation of these results as globally on a number of measures, the UAE is ranked as one of the safest countries in the world with the global Gallup Index ranking the UAE in the top 10 safest countries in the world [50,51]. This may have an influence on the extent to which youth in the UAE perceive how SM impacts their perception of national security. The findings in the study indicate a trend for SM to have a higher level of influence on the political dimension followed in decreasing order by Economic; Cultural and Societal; Ethical and Religious, with as indicated previously, the least influence being on the perception of security implications.

The results of the one-way analysis of variance indicated that the perception of the level of national security as it relates to youth in the UAE has the potential to be predicted through the impact of social media. The study revealed that the values of the regression coefficient and the beta coefficient (B) were positive in two dimensions, cultural/societal implications and economic implications, which indicates that these two dimensions are

directly proportional to the level of national security. Therefore, a change of 1% in the level of these two dimensions causes a change in the perceived level of national security by 36.6% and 6.9%, respectively. The values of the regression coefficient and beta coefficient (B) were negative in two dimensions: moral/religious implications and security implications, indicating these two dimensions negatively affect the perception of the level of national security. The study contribution to the wider community responsible for national security is multi-dimensional, including contributing to the earlier work in this field [9,10,17,18], and identifying the 'power' of social media to influence the perspectives of cultural, social, political, security, economic, and ethical/religious aspects of peoples' lives. Further, offering validation of the role of social media in the level of influence experienced by a sample of Arabic youth on five core elements is fundamental to national and international communities.

## 6. Conclusions

Social media is a tool of the 21st century utilized across the globe, with, as discussed previously, positive and negative implications. In respect of this study, the key areas of investigation were to: (a) develop an understanding from the perspective of UAE youth, the influence of social media on culture, economics, politics, religion, and security; and (b) investigate the potential for the perceived level of national security to be identified through social media from the perspective of UAE youth. The contribution of this study to the wider community of educators, scholars, researchers, and policy and decision makers associated with national security is offered through addressing the following aspects, i.e., theoretical contribution, practical implications and limitations, and future work.

### 6.1. Theoretical Contribution

This study contributes to the body of knowledge associated with understanding the influence of social media, in particular as it pertains to the influence on perceived levels of national security from the perspective of a representative sample of UAE youth. Further, the analysis process in this study offers insight into the respective influence of social media on key national dimensions, i.e., culture, economy, religion, politics, and security, as perceived by a sample of UAE youth. These findings are helpful contributors to future governance policy decisions in respect of the access and use of social media platforms

### 6.2. Practical Implications

The replicable nature of the study offers agencies responsible for national security insight into the perception of youth as to the capacity for social media to influence their perspective on national security dimensions. The survey design, whilst customized for this study, in parallel expands the earlier work in understanding the influence of social media on societal dimensions in conjunction with the influence of the level of national security perceived by youth. As indicated in Section 3, the literature offers a powerful message as to the influencing strength of 21st century social media and harnessing this strength for national (and international security and crisis management) demands attention. Results of such studies offer a contribution to the development of policies and legislation for the use of social media platforms, chief amongst which is the alignment with crisis management (the first phase of which, as indicated earlier, is mitigation) towards a more secure global society. In addition, the results of this study signpost the importance of engaging youth in promoting the positive use of social media across core dimensions of modern life as an avenue to enhance the perceived sense of security within a community and the potential to contribute to crisis and disaster preparedness.

### 6.3. Limitations and Future Work

This study is a preliminary exploration of the influence of SM on a sample of Arab Youth in respect of perception of national security. The findings indicate that the perceived level of national security has the potential to be predicted through the influence of social

media. With the findings in this study indicating a residual greater than 81%, there is fertile ground to build on this baseline research study to explore the composition of residual elements. Importantly, the research design is replicable, and it would contribute to a balanced perspective of the influence of SM on perspectives of national security to apply the research in communities experiencing different levels of current economic and national security to those experienced in the UAE. A further area to explore a contrast study based on the textual analysis of opinions shared on social media has been undertaken in the work of Marine Roig [15–17], for example. Such a study inclusive of other western countries and delineating gender-based responses would extend the baseline work presented in this KSA study. Whilst acknowledging there are limitations to this research project, i.e., restricted to one Emirate, limited demographic details of the sample group (for example, gender, employment status, level of education, time allocated to SM), there is a myriad of avenues for further research in this field, utilizing this baseline study to expand the global understanding of social media on the views and opinions of the next generation of national and international leaders, the youth of 2022.

**Author Contributions:** Conceptualization, N.A.N.; methodology, Author N.A.N. and N.A.M.; software, N.A.N.; validation, N.A.N. and N.A.M.; formal analysis, N.A.N.; investigation, N.A.N.; resources, N.A.N.; data curation, N.A.N.; writing—original draft preparation, N.A.N., N.A.M. and A.D.; writing—review and editing, A.D.; visualization, A.D.; supervision, N.A.M.; project administration, N.A.M.; funding acquisition, not applicable. Author N.A.N. undertook the data collection, supervised by author N.A.M. Author A.D. developed and submitted the article based on the data collected and analyzed by N.A.N. All authors have read and agreed to the published version of the manuscript.

**Funding:** This research received no external funding.

**Institutional Review Board Statement:** The study was conducted in accordance with the Declaration of Helsinki, and approved by the Dubai Police Academy, Approval Code: DPA-2021-3-006-1 (third semester, Council meeting August 2021).

**Informed Consent Statement:** Informed consent was obtained with completion of the anonymous survey obtained from participants involved in the study.

**Data Availability Statement:** The data are not publicly available due to nature of the survey respondent's workplace and document security.

**Conflicts of Interest:** The authors declare no conflict of interest.

## Appendix A

| Survey form |
| :---: |
| **Social media and its implications for national security** |
| **(Applied study on the United Arab Emirates)** |

### After Greetings

I am pleased to put in your hands this questionnaire, which comes in the context of an effort to complete the requirements for obtaining a master's degree in security crisis management at the Dubai Police Academy, and which came under the title (**Social Media and Its Repercussions on National Security and Ways to Confront It—An Applied Study on the United Arab Emirates**). The questionnaire aims to stand on the level of national security in the UAE through the repercussions of social media on members of society, and to clarify the role of the relevant agencies to enhance national security in the UAE by facing the negative repercussions of social media.

Therefore, we hope for your cooperation by filling out the attached form with all honesty and objectivity towards all the items specified in the questionnaire's list of phrases, knowing that all information will be used only for the purposes and service of scientific research, and will be treated with complete confidentiality.

**First: general data**

**Put a tick (√) in front of your choice:**

| -1 | Gender | Male ( )<br>Female ( ) |
|----|--------|------------------------|
| -2 | Academic qualification | ( ) Diploma or less<br>( ) BA<br>( ) Postgraduate |
| -3 | Work nature | ( ) Responsible/Director<br>( ) employee<br>( ) does not work |
| -4 | Nationality | ( ) Emirati<br>( ) resident |
| -5 | Age | ( ) From 20 to 25 years old<br>( ) From 26 to 30 years old<br>( ) from 31 to 40 years |

**Second: The axes and paragraphs of the questionnaire**

**The first axis: the implications of social media on national security**

| M | Axis/Paragraph | I Totally Agree | I Agree | Neutral | Disagree | Strongly Disagree |
|---|----------------|-----------------|---------|---------|----------|-------------------|
| | **First—The cultural and societal repercussions of social media on the national security of the UAE** | | | | | |
| 1 | Social media helps spread its disruptive ideas in society, especially among young people | | | | | |
| 2 | Social media has helped spread Western ideas, values, and customs among young people in Emirati society | | | | | |
| 3 | Social media helps to be influenced by the opinions, beliefs and perceptions of others and to be followed by young people in the UAE | | | | | |
| 4 | Social media has helped form groups and friendships between young people with common cultural and scientific interests | | | | | |
| 5 | Social media has contributed to easy access to information and news in an instant and its repercussions on the Emirati society | | | | | |
| 6 | Social media plays a major role in questioning the value of the country's cultural heritage and national symbols | | | | | |

| M | Axis/Paragraph | I Totally Agree | I Agree | Neutral | Disagree | Strongly Disagree |
|---|---|---|---|---|---|---|
| **Second—The moral and religious repercussions of social media on the national security of the UAE** | | | | | | |
| 1 | Social media contributes to the fluctuation of the value system of members of the Emirati society as a result of the mixing of cultures | | | | | |
| 2 | Social media helps to spread the websites of deviant and misleading teams and groups | | | | | |
| 3 | Social media has contributed to the development of values associated with the moderation and tolerance of the Islamic religion in the United Arab Emirates | | | | | |
| 4 | The spread of social media has contributed to a gap between religious scholars and the youth group in society | | | | | |
| 5 | Social media has contributed to the spread of immoral and immoral images and videos among young people in society | | | | | |
| 6 | Social media has contributed to the marketing of consumer values that are hostile to our authentic Arab values, ethics and customs | | | | | |
| 7 | Social media lacked depth in personal relationships between members of the same family, and the moral aspect of family control was weak | | | | | |
| **Third—The political repercussions of social media on the national security of the UAE** | | | | | | |
| 1 | Social media contributes to mobilizing public opinion and news against government policy in the United Arab Emirates | | | | | |
| 2 | Social media distorts the personal information of some important leadership figures in the United Arab Emirates | | | | | |
| 3 | Social media helps to respond to and refute the suspicions raised about the UAE | | | | | |
| 4 | Social media develops the ability of young people in the UAE to objectively express their point of view on community issues | | | | | |
| 5 | Social media contributes to promoting the values associated with the concepts of citizenship and national responsibility | | | | | |

| M | Axis/Paragraph | I Totally Agree | I Agree | Neutral | Disagree | Strongly Disagree |
|---|---|---|---|---|---|---|
| **Fourth—The economic repercussions of social media on the national security of the UAE** | | | | | | |
| 1 | Social media has contributed to the increase in money laundering | | | | | |
| 2 | Social media is a business investment that benefits business owners | | | | | |
| 3 | Social media negatively affects the country's economy | | | | | |
| 4 | Social media has contributed to the spread of illegal e-marketing | | | | | |
| 5 | Social media provides a suitable environment for e-commerce buying and selling | | | | | |
| 6 | Social media contains annoying and often unacceptable advertisements | | | | | |
| **Fifth—The security implications of social media on the national security of the UAE** | | | | | | |
| 1 | Social media contributes to the formation of public opinion towards security issues no matter how valid they are | | | | | |
| 2 | Social media has indirectly violated a lot of others' privacy through spying and malicious software | | | | | |
| 3 | Social media has a role in directing false information and news that may lead to crises and security disturbances | | | | | |
| 4 | Social media contributes to spreading hate crimes in light of the multinationality of the United Arab Emirates | | | | | |
| 5 | Social media has helped increase the rate of cyber-extortion crimes in Emirati society | | | | | |
| 6 | Social media contributes to spreading rumors that harm the security and stability of society in the United Arab Emirates | | | | | |

| | The second axis: the level of national security in the UAE through the repercussions and effects of social media on society from the point of view of the youth group. |
|---|---|
| 1 | The wide spread of social media represents a real threat to the national security of the United Arab Emirates |
| 2 | The contents published on social media pages are considered an entry point for the moral, religious and cultural invasion of young people in the UAE |
| 3 | Social media contributes to reducing opportunities for interaction and communication between family members in the UAE, which affects the level of national security |
| 4 | Social media constitutes a fertile environment for some to exploit extremist ideas that affect the security and stability of Emirati society |
| 5 | Social media has helped young people in the UAE learn about and benefit from other cultures and civilizations |
| 6 | The competent security agencies in the UAE tightly control the contents of social media that are harmful to the country's national security to limit its political and security impact on members of society |
| 7 | Social media is used as media platforms by the youth group in the UAE to spread the values of tolerance in light of the country's multiple cultures |
| 8 | Developed legislation and laws have contributed to limiting the security and social effects of social media on the national security of the state |
| 9 | There is intellectual, cognitive and cultural maturity among young people in the UAE with the threats of social media to national security and ways to deal with them |
| 10 | The security media of the Ministry of Interior and police leaders contributed to spreading security awareness among members of society about the effects of social media on the national security of the state and its negative repercussions on stability |

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
