# Peer review of "The Influence of Social Media on Perceived Levels of National Security and Crisis: A Case Study of Youth in the United Arab Emirates"

_sustainability, doi:10.3390/su141710785_

Round 1

Reviewer 1 Report

The influence of social media on national security and crisis:

A case study of youth in the United Arab Emirates

The topic is interesting and the sample is considerable, but the structure and format of the manuscript are not correct. In addition, it is necessary to delve into the impact of security or safety, reflected in social media, on the online image of the country, compared to the influence of the national and international media (press, radio, and television) in this matter. Here are some recommendations to improve the manuscript structure and content.

Abstract:

“The results of the study on the effects and consequences of social media on national security in the UAE,”

“National security” is an indeterminate concept. It is more accurate to speak of the perceived level of national security, and the country image projected by social media and perceived by young people in relation to national security.

Also, in the title and abstract, you speak of “security” and, in the text, of “safety and security”. Since some authors consider both concepts synonymous, you should define them in the text for the purposes of this study.

Keywords: Three to ten pertinent keywords need to be added after the abstract. We recommend that the keywords are specific to the article, yet reasonably common within the subject discipline.

To increase the visibility of the article through bibliographic search engines, you could add other keywords such as: multiple linear regression; Emirate of Sharjah; United Arab Emirates

Background

The literature review is important enough to constitute a first level section. Therefore, you should add the section 2. Background to contain the current subsection 1.1.

At the end of the Introduction section, you must include the rationale of the study, that is, the aims or objectives of the research, or the questions that the research intends to answer. For example, the content of the paragraph between lines 174-186 should be at the end of the Introduction section.

At the end of the new section 2. Background, you must include a diagram with the constructs you want to analyse and their relationships, so that readers can see graphically what the conceptual model is. The division that you have made between Category One (implications) and Two is confusing. The constructs and their relationships must be based on previous research.

Given that issues related to crises are disseminated mainly through the media (press, radio and television), in this background section, it is necessary to delve into the presence of security issues in social media. Please see (ResearchGate): “How safety affects destination image projected through online travel reviews” (JDMM, 2020), on Airbnb guest reviews, and “Content analysis of online travel reviews” (Springer, 2022), on TripAdvisor visitor reviews, because both publications deal with the social media presence of a terrorist attack followed by a secession movement in a Spanish region.

3. Materials and Methods

Please review the wording of the manuscript. For example: “their total number is approximately (19,600),” why do you put 19,600 in parentheses if it is a direct part of the sentence?

The constructs and items of the questionnaire are unclear. Both must be based on previous research. Please include the questionnaire translated into English in an annex and dedicate a column to citation of previous reliable research related to the constructs and items.

4. Results

The space occupied by the tables is not optimised. Please see Table 2 of the

sustainability-template.dot document.

5. Discussion

Both in the Background section and in this section, you should also comment on the results of other related studies.

6. Conclusion

It is necessary to delve into the implications of the study. Concluding remarks section should have a summary of the main study outcomes and the way to obtain them, and three subsections:

6.1. Theoretical implications or Theoretical contribution: Does the study contribute to the body of knowledge on national safety or security from a theoretical perspective?

6.2. Practical implications: Can the study be useful for academics, security forces or social media users?

6.3. Limitations and future work: A simple study cannot cover all aspects of a subject as complex and extensive such as the impact of social media on the projection / perception of a country's security or safety. What could be the future lines of research in this area? For example, a study focused on user opinions about safety or security in the country, shared online on social media (e.g., JDMM, 2020; Springer, 2022).

Reference list

Please check the citations and references: Multidisciplinary Digital Publishing Institute (MDPI) style. I use Mendeley.com or Zotero.org with the MDPI template.

For example: Abbreviated Journal Name

Author Response

Thank you sincerely to Reviewer # 1 for the valuable and insightful comments and recommendations. Please kindly find attached the Authors' response. 

Reviewer 2 Report

This study explores the perception of United Arab Emirate community members (specifically youth) on the influence of social media as a threat the wide use of SM platforms for Emirate of Sharjah (Dibba Al-Hisn, Khor Fakkan, Kalba) were analyzed utilizing a descriptive-analytical method.

Please consider the following points for revision:

1.     Authors should clarify the contribution in the introduction. In the literary review, include a comparative table with the different papers and highlight what the contribution of the proposed work would be.

2.     Section 2. In the data collection, the authors should better explain the data collection. What were the questions? And they need to justify the choice of the questionnaire.  The format of the answers is not clear. This section needs to improve a lot.

3.     The proposed results and analyzes are too basic. The authors need to include other methodologies (eg multiple correspondence analysis, clustering or others.).

4.     Conclusion and Future work are given in brief. Detail discussion on existing work, limitations of existing work, proposed method and future work should be added in this section.

Author Response

thank you most sincerely for the time and effort taken to offer recommendations to improve the standard of the submission. 

Reviewer 3 Report

All around an interesting paper. I would appreciate it if the authors put more effort into presenting similar previous research in UAE or worldwid

Author Response

Thank you for the supportive comments, recommendations related to literature have been incorporated.

Please see attached 

Round 2

Reviewer 1 Report

The influence of social media on perceived levels of national security and crisis:

A case study of youth in the United Arab Emirates

The manuscript has improved significantly and I recommend its publication in Sustainability once minor changes have been addressed.

There is a crucial aspect about the influence of social media that I do not see reflected in the manuscript. In section 1. Introduction, it is necessary to inform the readers that the content generated by users (UGC) and shared on social media is an agent of formation or source of information of the online country image, as it was demonstrated in “Destination image analytics through traveller-generated content” (Sust., 2019). Likewise, this study highlighted the subjectivity of the perceived image. In relation to this case study, surely the results would be different if the survey was distributed among older inhabitants, or among young foreigners, especially from Western countries. A related study found safety or security to be an important attribute of country image; please see “How safety affects destination image projected through online travel reviews” (JDMM, 2020).

In the “Limitations and Future Work” section, a contrast study based on the textual analysis of opinions shared on social media (UGC) needs to be added. Please see Sust. (2019) and JDMM (2020).

Figure 1: Please widen the figure to optimise the space occupied (see Figure 2 of the document: sustainability-template.dot).

Table 1: Please follow the MDPI style (see Table 2 of the document: sustainability-template.dot).

Good luck!

Author Response

Dear Reviewer

thank you very much for all of your guidance, as there were a number of changes when combining the two reviewer recommendations, the changes are shown in highlight yellow 

with regards 

Reviewer 2 Report

The authors say "The study sample

consisted of (1185) young people (male/female) who were chosen by a simple random method" which method?why not use the total of the data set?

how many are male and female? maybe the sample is unbalanced

"Thank you for your comments. Unfortunately, it is not possible to re- access the raw data due to unforeseen  circumstances beyond the authors' control". This is very strange, the authors or other researchers could not replicate the study?

Author Response

Dear Reviewer

thank you sincerely for all of your guideance and patience in supporting our development of the article. as there were a number of changes when combining the reviewer 1 and 2 comments, the changes have been highlighted in yellow for ease or readability vs track changes. 

with regards 

Round 3

Reviewer 2 Report

I have no more comments.

This manuscript is a resubmission of an earlier submission. The following is a list of the peer review reports and author responses from that submission.